# The Effects of *Origanum vulgare* L. Essential Oils on Anaesthesia and Haemato-Biochemical Parameters in Mozambique Tilapia (*Oreochromis mossambicus*) Post-Juveniles

Ndakalimwe Naftal Gabriel [1,*], Gadaffi M. Liswaniso [2], Wilhelm Haihambo [1] and Kenneth Prudence Abasubong [3]

[1] Department of Fisheries and Ocean Sciences, Sam Nujoma Campus, University of Namibia, Private Bag 462, Henties Bay 9000, Namibia
[2] Sam Nujoma Marine and Coastal Resources Research Centre, Sam Nujoma Campus, University of Namibia, Henties Bay 9000, Namibia
[3] Key Laboratory of Aquatic Nutrition and Feed Science of Jiangsu Province, College of Animal Science and Technology, Nanjing Agricultural University, Nanjing 210095, China
* Correspondence: ngabriel@unam.na or gnaphtal85@gmail.com

**Abstract:** This study investigated the effects of oregano (*Origanum vulgare*) essential oil (OEO) anaesthesia and stress-related physiological parameters in tilapia mossambicus (*Oreochromis mossambicus*) post-juveniles. Fish were subjected to different concentrations of OEO (25, 50, 100, and 150 μL L$^{-1}$) to assess the effects of the anaesthesia and recovery time. A second experiment subjected fish to other handling treatments, including a control (no OEO exposure), ethanol solution, 25 μL L$^{-1}$ (low effective anaesthesia concentration of OEO), and 100 μL L$^{-1}$ high effective anaesthesia concentration of OEO) for 10 min to assess the haemato-biochemical indices and survival rate at 0 h and after 24 h. Moreover, all the tested OEO concentrations induced anaesthesia in the studied fish, where the anaesthesia induction time decreased with increased OEO concentrations. Meanwhile, the inverse was reported for recovery time. Based on the ideal anaesthetic criteria, 50 μL L$^{-1}$ and 100 μL L$^{-1}$ were the suitable concentrations that could be recommended for quick anaesthesia. However, according to the haemato-biochemical parameters and survival results, 25 μL L$^{-1}$ was safe to anaesthetise tilapia mossambicus for 10 min and could be recommended for time-consuming fish-handling procedures. Future studies should investigate multiple factors that influence anaesthesia in fish for better optimisation of OEO in tilapia mossambicus.

**Keywords:** aquaculture; animal welfare; essential oils; herbal anaesthetics; tilapia mossambicus



## 1. Introduction

The aquaculture sector is relatively young in Namibia and an under-exploited food-producing industry, with the potential to empower ordinary persons to play a part in reducing food insecurity in the country. Even though the Namibian government supports aquaculture development (No. 18, 2002) [1], there are limited guidelines on the codes of practice regarding handling farmed fish throughout the production process, not to mention the biosecurity measures. The lack of care for fish welfare seems to be a global aquaculture concern, not simply a Namibian one. In Brazil, the humane slaughter regulations do not include fish [2], despite the call from the World Organization for Animal Health-OIE to recognise fish welfare during aquaculture processes (i.e., sampling, spawning, harvest) [3].

Namibia and other countries could learn from countries such as Norway and Sweden, which have guidelines and regulations that address fish handling practices during culture and capture operations [4]. This is crucial because aquaculture production and fishing activities, such as harvesting, transportation, and artificial spawning, can be stressful to the fish if they are not handled appropriately. This violates their welfare and may be considered unethical. As a result, fish are routinely anaesthetised during handling to decrease stress,

prevent harm, and protect fish welfare [5]. This could be done by using synthetic chemicals, such as tricaine methanesulphonate (MS-222) [6], 2-phenoxyethanol [7], and etomidate [8]. Synthetic anaesthetics are effective, but they are in short supply, particularly in developing countries; are often prohibitively expensive; and/or may have a deleterious effect on fish welfare (high stress and haematological disorders) [9]. As a result, the aquaculture and fishing sectors need safer and longer-lasting fish anaesthetics.

The focus of recent research has been on the use of herbal extracts, including essential oils (EOs) and their compounds, such as carvone, myrcene, linalool, menthol, cineole, and eugenol, as anaesthetics in fish [10,11]. These extracts are a superior substitute because they are effective; have minimal side effects; are affordable; are readily available locally; and have additional benefits, such as antioxidant, stress-relieving, antimicrobial, and immune-stimulating properties [11]. For instance, the plant essential oil derivatives myrcene [10] and linalool [10,12] were shown to induce anaesthesia in common carp (*Cyprinus carpio*). Similarly, the anaesthetic effectiveness of *Aloysia triphylla* [13], *Ocinum americanum*, and *Lippia alba* [14] was confirmed in *Oreochromis niloticus*, *Origanum* sp. and *Eucalyptus* sp. were effective in *Dicentrarchus labrax* and *Argyrosomus regius* [5], and *Aniba rosaedora* was effective in goldfish [15]. This suggests that plant EOs have the potential to replace, reduce the use of, or offer a substitute for synthetic anaesthetics in aquaculture and fisheries. Furthermore, given that many are easily grown, inexpensive, simple to extract, and widely accessible, they may be suitable for Namibia and other developing countries. However, more studies are required on the effects of EOs on various fish species, sizes, and environmental factors before they are used as anaesthetics in aquaculture [16]. This is so because the effects of anaesthesia can depend on many factors.

Oregano is one of the medicinal plants that are rich in EOs. Its EOs were reported in aquaculture as a potential feed additive in Nile tilapia (*Oreochromis niloticus*) [17] and as an effective anaesthetic in European sea bass (*Dicentrarchus labrax*), meagre (*Argyrosomus regius*) juveniles, and in silver kob (*Argyrosomonus inodorus*) adult fish [5,18,19]. The generally observed trend is that EOs induce anaesthesia in a concentration-dependent manner, with the anaesthesia intensity increasing with increased concentration. Oregano EO was found to be effective at inducing anaesthesia in *Dicentrarchus labrax* at concentrations ranging from 50 to 75 μL L$^{-1}$, with 50 μL L$^{-1}$ being determined to be the optimal concentration because it could reduce stress in fish after anaesthesia [18]. The effective concentration in *Argyrosomonus inodorus*, however, ranged from 25 to 50 μL L$^{-1}$. To date, the anaesthetic assessment of oregano is still limited in aquaculture, especially in freshwater fish species. To our knowledge, no prior study investigated the anaesthetic effects of oregano in tilapia mossambicus (*Oreochromis mossambicus*) which is one of the significant freshwater aquaculture species in Namibia. Therefore, this study investigated the effects of oregano (*Origanum vulgare*) essential oil's (OEO's) anaesthesia and stress-related physiological parameters in tilapia mossambicus (*Oreochromis mossambicus*) post-juveniles.

## 2. Materials and Methods

*Oreochromis mossambicus* juveniles (body weight range, 31.05–32.15 g) were bred and raised at the Sam Nujoma Marine and Coastal Resource Center (SANUMARC) freshwater research facilities, Sam Nujoma Campus, University of Namibia, Henties Bay, between November 2021 and March 2022. One hundred and twenty-six fish were acclimatised in a 500 L white round fibre glass tank filled with 300 L of dechlorinated and aerated freshwater (dissolved oxygen 5.21 ± 0.32 mg L$^{-1}$, pH 7.51 ± 0.13, and temperature 28.56 ± 0.76 °C), which was measured using a multi-parameter water quality meter (Eutech instruments, model PCD 650, part of Thermo Fisher Scientific, Singapore). To preserve fish welfare, they were maintained under a natural photoperiod (12 h of light and 12 h of darkness), and to maintain the water quality, one-third of the water was exchanged biweekly. Fish were fed ad libitum twice daily with a commercial diet (2 mm die, Aqua-Plus, Avi-Products Pty Ltd., Cato Ridge, KwaZulu-Natal Province, South Africa).

## 2.1. Plant Extracts and Chemical Analysis

Oregano EO (OEO) with 100% purity was obtained from a local natural products store (Klein Swakopmunder Distribution cc, Swakopmund, Namibia). The OEO was dissolved in absolute ethanol at a 1:10 ratio to enhance the emulsification before it was poured into the test water 10 min prior to fish anaesthetic exposure [20]. The OEO used in this study was the same as the one used in our previous study, and it contained the following main components: carvacrol (73.15%) r-cymene (5.32%), monoterpene (3.45%), g-terpene (3.08%), b-caryophyllene (2.5%), and thymol (2.32%) [19].

## 2.2. Anaesthetic Effectiveness Experiment

After the acclimation period, the fish were starved for 24 h prior to the anaesthesia and recovery experiment. A pilot study was performed to determine the experimental OEO concentrations. The concentrations were selected based on the criteria that define an ideal anaesthetic agent: the drug should induce anaesthesia within 180–300 s and recovery within 600 s [15]. The predetermined concentrations tested were as follows: 10, 25, 50, 100, 150, and 200 $\mu$L L$^{-1}$, and were chosen based on previous research [5]. Nine fish were randomly and individually exposed to each anaesthetic concentration, and the anaesthesia and recovery responses were observed according to Mylonas et al. [21] (Table 1). The final experimental concentrations were adjusted to 25, 50, 100, and 150 $\mu$L L$^{-1}$.

**Table 1.** Stages of anaesthesia (A) and recovery (R) in fish [21].

| | **Fish Behaviours/Response** |
|---|---|
| **Stages of anaesthesia** | |
| $A_I$ (initial induction) | Total loss of equilibrium, slow but regular opercular rate. |
| $A_D$ (deep anaesthesia) | No reflex, opercular movements slow and irregular, no response to strong external stimuli. |
| **Stages of recovery** | |
| $R_I$ | Partial regain of equilibrium, no active swimming. |
| $R_F$ | Full equilibrium regained, normal active swimming. |

After the preliminary test, nine fish were individually exposed to each concentration of OEO in a 5 L aquarium, and their anaesthetic and recovery stages were observed. The tests were carried out in ascending order (from lowest to highest concentration), with a complete water change after each replicate of each concentration, respectively. The water used to fill the 5 L experimental aquarium was from the acclimation tanks to ensure that the same conditions were provided during the experiment. Fish were only used once, with each fish being treated as a replicate [15,19]. When a fish attained deep anaesthesia ($A_D$), as described in Table 1, it was removed immediately from the anaesthesia aquarium. It was transferred to a fresh 5 L recovery aquarium until it attained full recovery ($R_F$) (Table 1). The time taken by each fish to reach each anaesthesia and recovery stage was recorded with a stopwatch by four researchers, with each researcher assigned a stage and with each stage allocated a maximum of 15 min for observation. Fish were monitored for mortality and abnormal behaviour three days after recovery.

## 2.3. Collection of Blood and Haemato-Biochemical Testing

After the anaesthesia experiment, the remaining fish (9 fish per group) were randomly divided into four groups: control (water only), ethanol bath (1500 $\mu$L L$^{-1}$ of ethanol, the volume used to dilute the highest concentration of OEO), and two OEO concentrations (25 $\mu$L L$^{-1}$ and 100 $\mu$L L$^{-1}$). The two OEO concentrations used in this experiment were selected based on criteria for promoting light and deep anaesthesia [15,22], as presented in the first anaesthetic experiment in this study. Similarly, the two OEO concentrations were first diluted in absolute alcohol at a 1:10 ratio before being gently mixed with water. The fish bathed for 10 min in each treatment tank. Immediately after 10 min, three fish were

randomly selected for blood collection, which took approximately 1 min. Fish in the control group were gently and randomly net-scooped from the tank and were blindfolded with a wet cloth to avoid stressing them. Then, blood was collected from the caudal vein of three randomly selected fish per treatment with a 3 mL sterile hypodermic syringe and cautiously transferred into sterile EDTA heparinised 4 mL tubes at room temperature. Blood samples were collected at different time intervals: zero (immediately after the 10 min bath in OEO) and 24 h. The blood collected was divided into two portions: (1) for haematological analysis and (2) for biochemical analysis.

Glucose was measured with an ACCU-CHECK® glucometer (Roche Diagnostic, Mannheim, Germany) [14,23], whereas TC and TG were analysed using the Mission cholesterol metre (Acon Laboratories, Inc., San Diego, CA, USA) [24]. After the blood parameters, the fish were monitored for mortality and abnormal behaviour for seven days after recovery. This study was done following the University of Namibia's research ethics policy and guidelines (ethical clearance no. AREC/021/2020) and the National Commission on Research, Science, and Technology (certificate no. RCIV00022018).

### 2.4. Statistical Analyses

Data were tested for normality using the D'Agostino–Person omnibus and Shapiro–Wilk tests. One-way analysis of variance (ANOVA) was used to study the anaesthetic effects (anaesthesia and recovery, presented as mean $\pm$ standard error) of OEO. Significant differences between OEO concentrations were further compared using Tukey's multiple comparison test ($p < 0.05$). These analyses were performed using GraphPad Prism version 6.01 (GraphPad Software Inc., San Diego, CA, USA).

## 3. Results

### 3.1. Anaesthetic Effectiveness

All tested OEO concentrations were able to induce anaesthesia in the studied fish, with significant differences observed between some concentrations ($p < 0.05$) (Figure 1). Moreover, the general observation was that normal fish behaviours, such as active swimming and operculum movement, were reduced with time spent in the anaesthetic solution. The time taken by the fish to reach partial induction or deep anaesthesia decreased with increasing OEO concentrations (Figure 1). When compared with 25 μL L$^{-1}$ (115.5 $\pm$ 9.14 s), fish exposed to 150 μL L$^{-1}$ (37.83 $\pm$ 5.92 s), 100 μL L$^{-1}$ (39.27 $\pm$ 8.42 s), and 50 μL L$^{-1}$ (55.48 $\pm$ 6.32 s) had significantly shorter partial induction times ($p < 0.05$). Similarly, deep anaesthesia lasted significantly longer in fish subjected to 25 μL L$^{-1}$ (321.71 $\pm$ 10.67 s) compared with those subjected to 150 μL L$^{-1}$ (54.31 $\pm$ 7.36 s), 100 μL L$^{-1}$ (65.49 $\pm$ 7.97 s), and 50 μL L$^{-1}$ (115.80 $\pm$ 6.50 s) ($p < 0.05$).

The oregano EO concentrations also had significant effects on the recovery time ($R_P$ and $R_F$) of *O. mossambicus*: the fish recovery time increased with an increase in OEO concentrations (Figure 1). The partial recovery time was significantly longer in the fish subjected to 150 μL L$^{-1}$ (552.89 $\pm$ 21.05 s) than in those subjected to 25 μL L$^{-1}$ (265.48 $\pm$ 19.27 s) and 50 μL L$^{-1}$ (315.04 $\pm$ 21.13 s) ($p < 0.05$). The same response pattern was observed in fish during full recovery, and the full recovery time ranged from 394.04 $\pm$ 23.09 s to 654.53 $\pm$ 19.14 s. Moreover, the regression analysis showed a negative relationship between deep anaesthesia and full recovery time (shorter deep anaesthesia time was associated with a long full recovery time) (Figure 1). No fish mortality was recorded during the experiment or three days post-exposure.

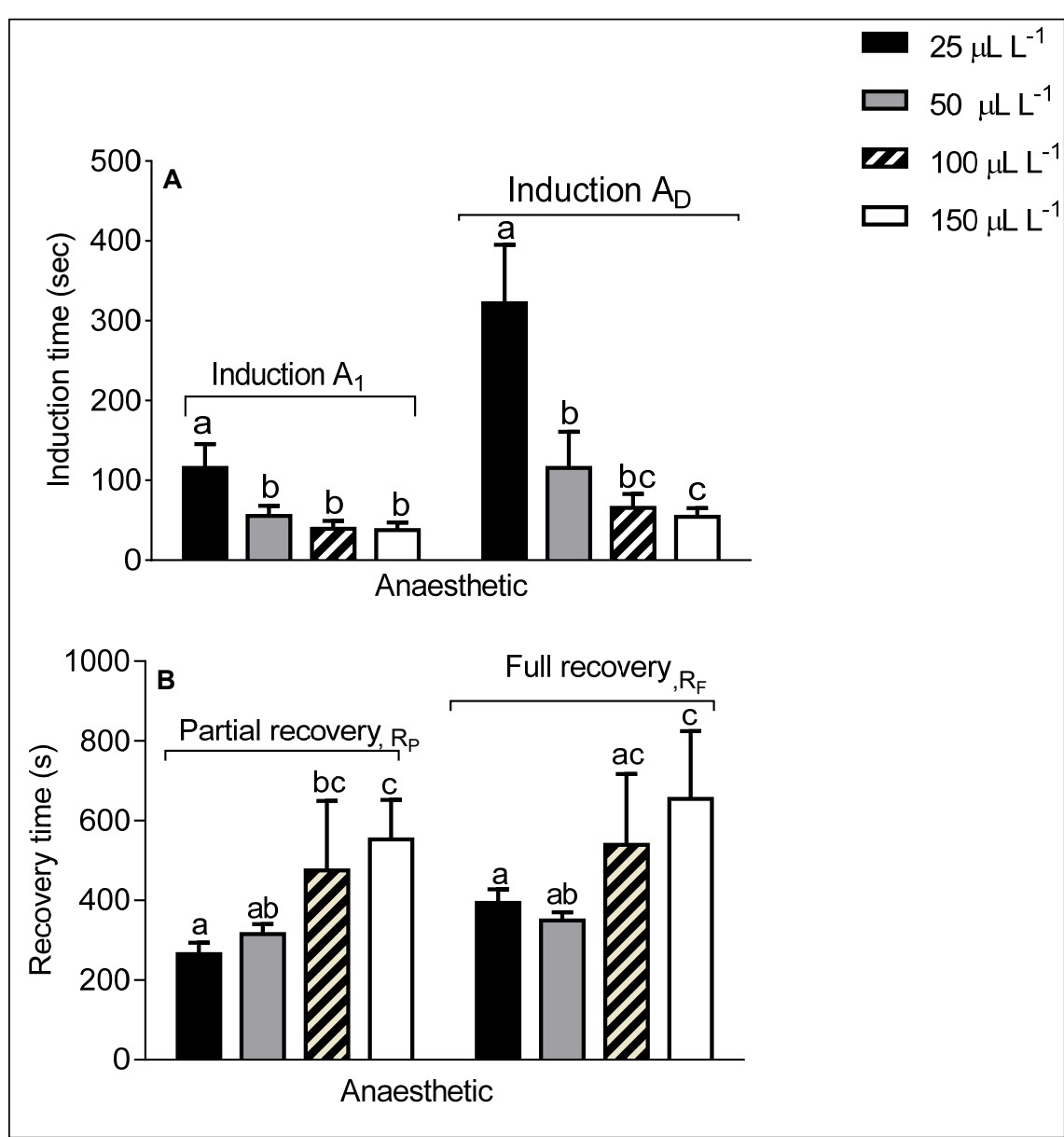

**Figure 1.** Induction ($A_1$ and $A_D$) (**A**) and recovery ($R_P$ and $R_F$) (**B**) time in Mozambique tilapia (*Oreochromis mossambicus*) subjected to different concentrations of oregano (OEO). Values are given as mean $\pm$ standard error ($N = 9$), and different lowercase letters indicate significant differences between treatment groups. Values were subjected to one-way ANOVA and Tukey's multiple comparison test to determine the significant differences between each anaesthetic stage and OEO concentrations at the $p < 0.05$ significance level.

### 3.2. Parameters of Biochemistry and Haematology

Generally, the fish haemato-biochemical parameters differed within and between treatment groups before and after 24 h (Figures 2–5). Fish exposed to 100 µL L$^{-1}$ had significantly higher glucose levels among the treatment groups before and after 24 h ($p < 0.05$), and this level was reduced significantly after 24 h ($p < 0.05$). On the other hand, total cholesterol and triglyceride levels did not differ significantly at 0 h and after 24 h compared with the control. Overall, the haematological parameters did not differ significantly between and within treatment groups after 24 h ($p > 0.05$). Among the treatment groups, fish mortality (44.44%) was only observed in fish bathed in 100 µL L$^{-1}$ of OEO during the seven days of monitoring.

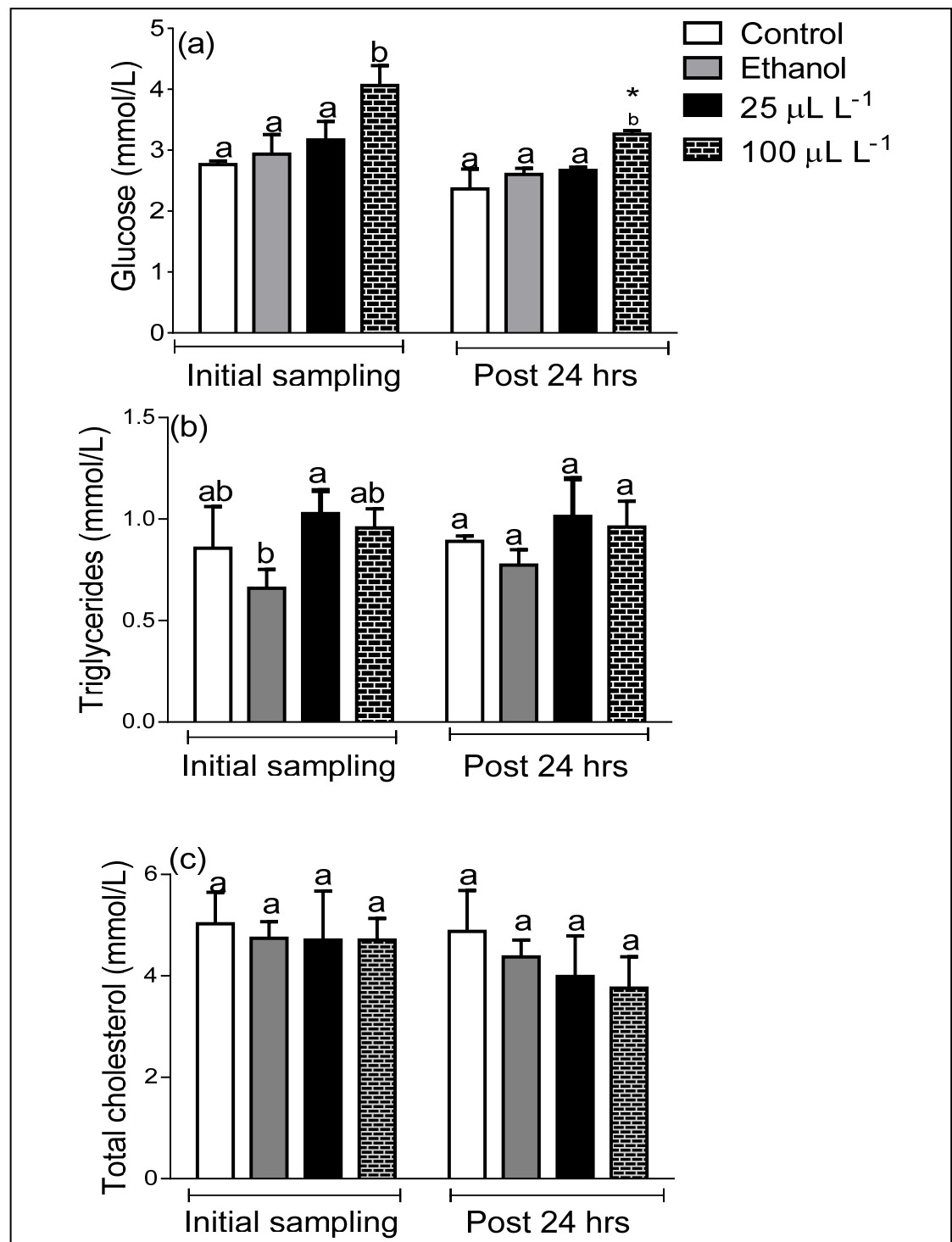

**Figure 2.** Blood glucose (**a**), triglycerides (**b**), and total cholesterol (**c**) of Mozambique tilapia (*Oreochromis mossambicus*) subjected to two oregano concentrations (25 and 100 µL L$^{-1}$), ethanol, and a control (water only). These parameters were sampled at 0 h and after 24 h. Values are given as mean ± standard error (*N* = 3), and different lowercase letters indicate significant differences between treatment groups within a sampling time (0 h or 24 h) using one-way ANOVA and Tukey's multiple comparison test. The asterisks indicate a significant difference of the treatment group from other groups after 24 h ($p < 0.05$).

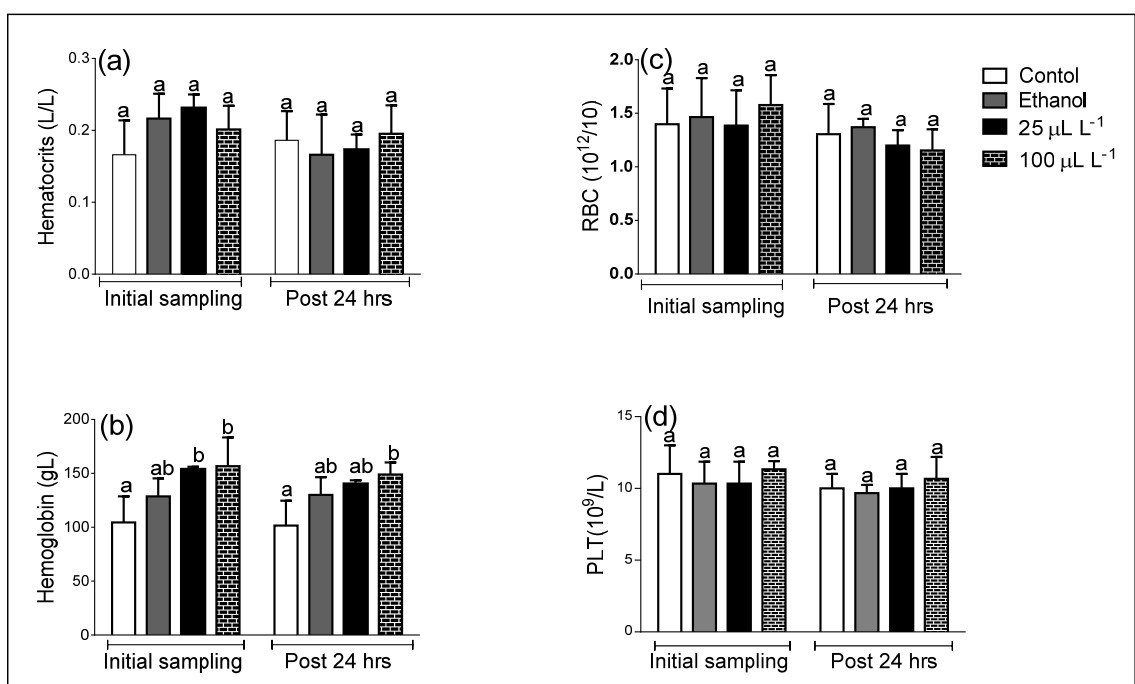

**Figure 3.** Hematocrits (**a**), hemoglobin (**b**), RBC (**c**), and PLT (**d**) of Mozambique tilapia (*Oreochromis mossambicus*) subjected to two oregano concentrations (25 and 100 µL L$^{-1}$), ethanol, and a control (water only). These parameters were sampled at 0 h and after 24 h. Values are given as mean ± standard error (*N* = 3), and different lowercase letters indicate significant differences between treatment groups within a sampling time (0 h or 24 h) using one-way ANOVA and Tukey's multiple comparison test.

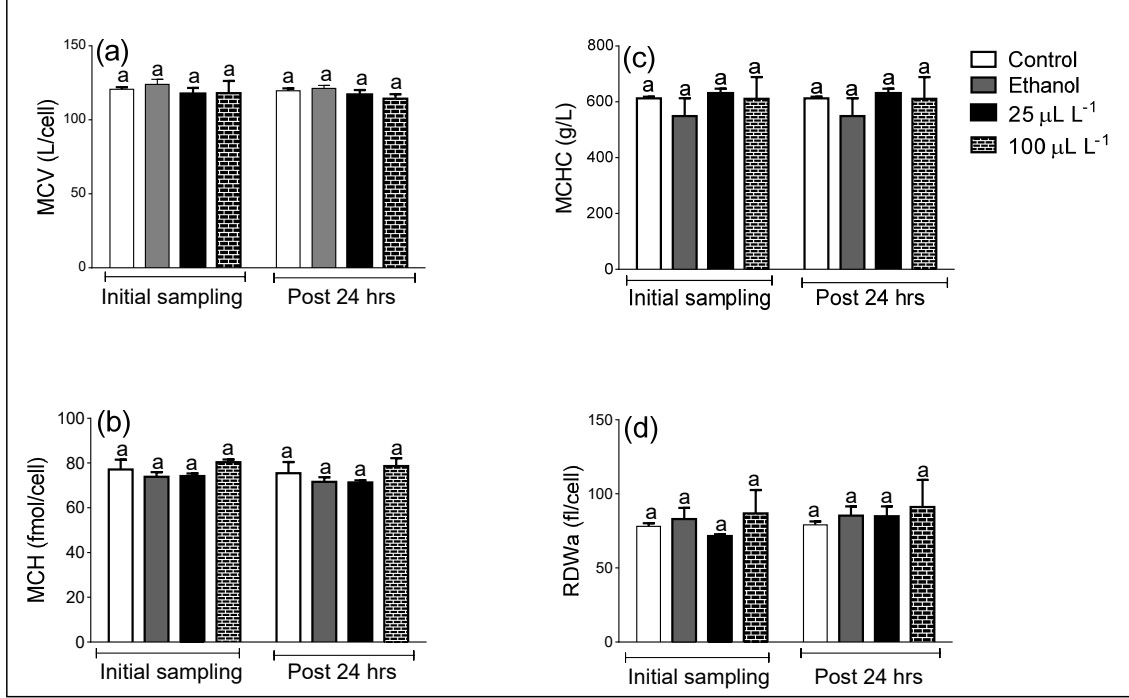

**Figure 4.** MCV (**a**), MCH (**b**), MCHC (**c**), and RDW$_a$ (**d**) of Mozambique tilapia (*Oreochromis mossambicus*) subjected to two oregano concentrations (25 and 100 µL L$^{-1}$), ethanol, and a control (water only). These parameters were sampled at 0 h and after 24 h. Values are given as mean ± standard error (*N* = 3), and different lowercase letters indicate significant differences between treatment groups within a sampling time (0 h or 24 h) using one-way ANOVA and Tukey's multiple comparison test.

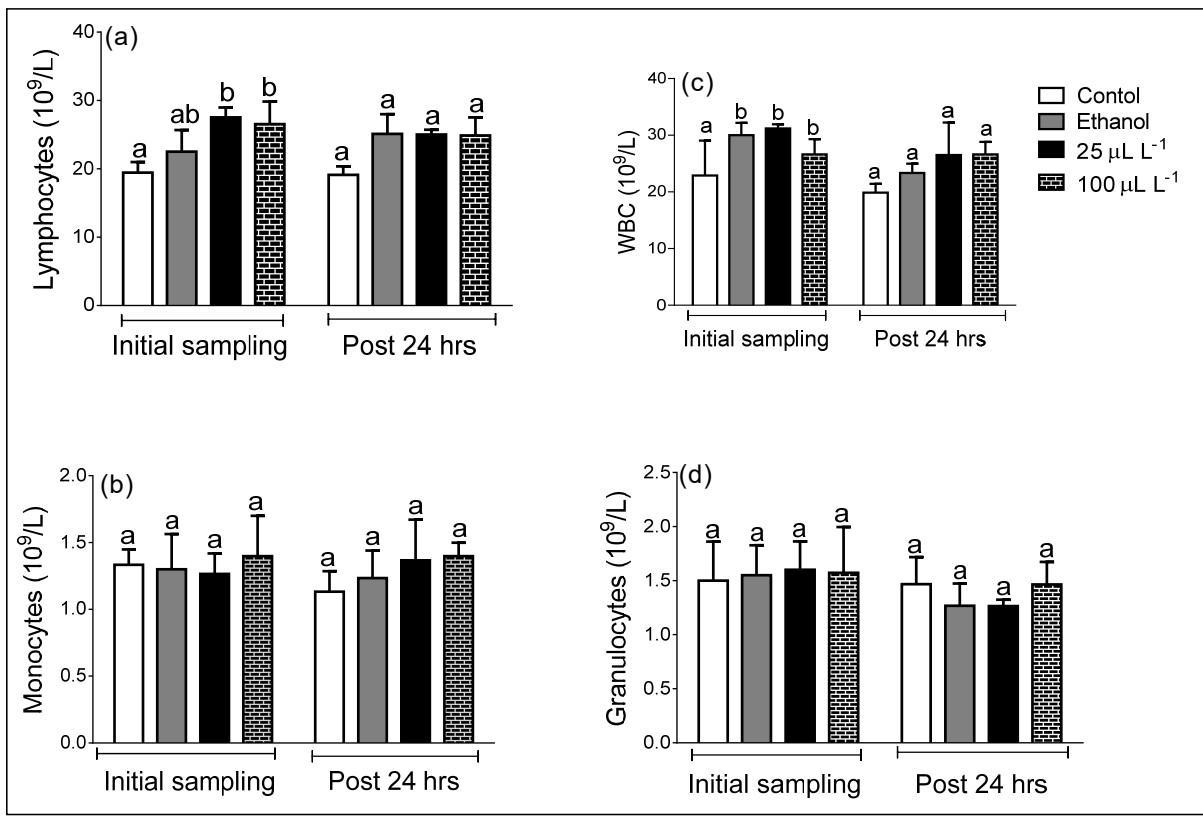

**Figure 5.** Lymphocytes (**a**), monocytes (**b**), WBC (**c**), and granulocytes (**d**) of Mozambique tilapia (*Oreochromis mossambicus*) subjected to two oregano concentrations (25 and 100 µL L$^{-1}$), ethanol, and a control (water only), respectively. These parameters were sampled at 0 h and after 24 h. Values are given as mean ± standard error (*N* = 3), and different lowercase letters indicate significant differences between treatment groups within a sampling time (0 h or 24 h) using one-way ANOVA and Tukey's multiple comparison test.

## 4. Discussion

Oregano essential oil has been mainly studied as a feed supplement in aquaculture [25–28]. It was first reported as a potential anaesthetic agent in two marine fish species, namely, European seabass [5,18] and meagre [18], and in doctor fish (*Garra rufa*) [29] and *A. inodorus* [19]. Therefore, to the best of our knowledge, this is the first study to investigate the anaesthetic effects of OEO in *O. mossambicus*. Carvacrol was the main compound in OEO (78.163%) [5,18,20]. Carvacrol is known as the chemotype, which is the compound that chemically distinguishes a plant. Its presence in the EO is far more important than its quantity [30]. This is due to the fact that the amount of it in the EO can vary depending on factors such as the plant's origin (i.e., climate, soil type, and water), how it was handled after harvest, and how it was extracted [31].

Previous studies showed that OEO or its chemotype extract (carvacrol) induced an anaesthesia response in the studied fish. Bodur et al. [5] exposed European bass (90.91 g) and meagre (127.43 g) to different concentrations of OEO (25, 50, and 75 µL L$^{-1}$), and discovered that 25 and 50 µL L$^{-1}$ concentrations could optimally induce an anaesthesia response in these fish. Aydin and Orhan recommended that 50 µL L$^{-1}$ of OEO carvacrol concentration could effectively induce anaesthesia in *Garra rufa* fish [29]. Similarly, the current study discovered that OEO effectively induced anaesthesia in tilapia mossambicus, with no mortality observed during the experiment or three days after recovery. Anaesthesia stages were observed in a distinct manner; this may indicate a broad safety limit for OEO as a potential fish anaesthetic, making it easier to apply. In addition, as the OEO concentration increased, the induction times decreased. This was inversely related to recovery times,

as demonstrated in various studies [5,13,29,32]. Studies on different pharmaceutical fish anaesthetics demonstrated that a specific concentration in the blood is required to effectively induce anaesthesia in fish [33]. This could be the case with EO; hence, studies such as the current study are critical in optimising EO as fish anaesthetics in aquaculture.

Furthermore, an ideal fish anaesthetic should induce deep anaesthesia (total loss of equilibrium, no reflex, and no response to strong external stimuli) in less than 180 s or 300 s [34,35]. It should also allow for rapid recovery in less than 600 s and leave little or no residues in the tissues [34,35]. In the current study, all of the tested concentrations of OEO could induce anaesthesia in *O. mossambicus*. However, only 50 $\mu$L L$^{-1}$ and 100 $\mu$L L$^{-1}$ of OEO met the criteria for an ideal anaesthetic in aquaculture. Although 50 $\mu$L L$^{-1}$ and 100 $\mu$L L$^{-1}$ of OEO seemed to be ideal concentrations, 25 $\mu$L L$^{-1}$ appeared to be safer (100% survival after seven days) than 100 $\mu$L L$^{-1}$ of OEO, which resulted in high mortality days after the welfare experiment (10 min OEO exposure). Similarly, high mortality was reported after silver catfish (*Rhamdia quelen*) were anaesthetised with carvacrol; however, fish mortality decreased as carvacrol concentrations increased [36]. Based on the findings of both the previous study and the current study, OEO (50 $\mu$L L$^{-1}$ and 100 $\mu$L L$^{-1}$) should only be used for quick procedures, such as biometric measurements and blood sampling in tilapia mossambicus. On the other hand, for lengthy procedures, such as artificial breeding, 25 $\mu$L L$^{-1}$ of OEO could be recommended.

Furthermore, the anaesthetic mechanism of EOs and their derivatives appears to be fairly understood at this point. According to Manayi et al. [37], essential oils induce anaesthesia due to their lipophilic nature, which allows them to penetrate through the cell membranes easily and influence brain functions by interacting with the gamma-aminobutyric acid (GABA) receptor, where GABA is an inhibitory neurotransmitter [37,38]. In other words, EOs may influence the swimming behaviour and consciousness of fish by increasing the activity of the GABA receptor, as demonstrated in *R. quelen* after exposure to carvacrol and thymol [36]. However, more research is needed to determine how different EOs and their chemotypes induce anaesthesia in different fish species.

In aquaculture, haemato-biochemical parameters are routinely used to assess fish health and welfare, and essential oils were reported to alter some haemato-biochemical parameters in fish. For example, immersing tropical pacu fish (*Piaractus mesopotamicus*) in *Lippia sidoides* EO for 10 min significantly increased haematocrit and glucose levels after exposure [30]. Some haemato-biochemical parameters were altered after *Ocimum basilicum* EO anaesthesia was also reported in tambaqui (*Colossoma macropomum*) fish [39]. Similarly, the current study found that the blood glucose levels were significantly higher in fish anaesthetised with 100 $\mu$L L$^{-1}$ of OEO. Even though this level significantly decreased 24 h after recovery, it remained significantly higher between treatments. The hyperglycaemic observation in fish post-recovery in the current study is a typical stress response and a physiological effort by an animal to counteract or survive stressful conditions [40]. This could be explained by increased fish activity, such as swimming and/or insufficient oxygen due to hypoventilation during post-anaesthesia recovery [41]. These conditions may be more severe in fish exposed to higher EO concentrations, as observed in the present study. Tambaqui [39] and pacu fish [30] exposed to *L. sidoides* and *O. basilicum* EO showed similar effects. In contrast, higher stress was reported in rainbow trout (*Oncorhynchus mykiss*) [42] bathed in 1,8-cineole solution and in Nile tilapia exposed to thymol solution [32]. This is believed to be due to prolonged anaesthesia associated with a lower EO concentration [32]. Hence, more research is required to determine how EOs and their compounds affect fish health and welfare post-anaesthesia.

Furthermore, as discussed above, the increased haemoglobin, lymphocytes, and WBC levels in fish exposed to 25 $\mu$L L$^{-1}$ and 100 $\mu$L L$^{-1}$ OEO post-recovery in the present study are a physiological response to increasing the oxygen capacity and resistance against hypoventilation stress post-anaesthesia. However, these parameters returned to their normal levels 24 h after anaesthesia. The same findings were reported in tambaqui [39]. In addition, regardless of the physiological stress observed in fish after EO anaesthesia,

lipids (total cholesterol and triglycerides) were not affected in the present study and the study of Ventura et al. [39].

In conclusion, in this study, OEO induced anaesthesia in Mozambique tilapia post-juveniles in a concentration-dependent mode, with 50 µL L$^{-1}$ and 100 µL L$^{-1}$ being the ideal anaesthetic concentrations that can be recommended for quick anaesthesia. However, from a welfare standpoint, only 25 µL L$^{-1}$ was safe to anaesthetise Mozambique tilapia for 10 min. Hence, it can be recommended for time-consuming procedures. It was been demonstrated that diluting EOs in ethanol had no significant effect on fish welfare. Furthermore, since the present study was limited to post-juvenile Mozambique tilapia, additional studies are needed to understand the anaesthetic and physiological effects of OEO in different sizes of Mozambique tilapia, as well as replication in other aquaculture species.

**Author Contributions:** Conceptualization, N.N.G., G.M.L.; methodology, N.N.G. and G.M.L.; a formal analysis and investigation, N.N.G., G.M.L. and W.H.; writing original draft preparation, N.N.G.; writing—review and editing, N.N.G., G.M.L. and K.P.A.; visualization, N.N.G.; project administration, N.N.G. All authors have read and agreed to the published version of the manuscript.

**Funding:** This study had financial support from UNAM Research and Publications Committee (URPC) Research Grant URPC/2015/277 and UNAM Pro-Vice Chancellor for Research, Innovation, and Development (PVC: RID) Research Grant no. 4515/2901.

**Institutional Review Board Statement:** This study was done following the University of Namibia's research ethics policy and guidelines (ethical clearance no. AREC/021/2020) and the National Commission on Research, Science, and Technology (certificate no. RCIV00022018). To prevent fish stress and harm, blood samples were collected only by an experienced researcher, and a reasonable number of fish were used in the process.

**Informed Consent Statement:** Not applicable.

**Data Availability Statement:** The data presented in this study is available on request from the corresponding author.

**Conflicts of Interest:** The authors declare no conflict of interest.

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
