# Peer review of "The Effects of Origanum vulgare L. Essential Oils on Anaesthesia and Haemato-Biochemical Parameters in Mozambique Tilapia (Oreochromis mossambicus) Post-Juveniles"

_2673-9496, doi:10.3390/aquacj2040015_

Round 1

Reviewer 1 Report

Comments are in attached document

Reviewer 2 Report

The current study investigated the effectiveness and side effects of oregano essential oils (OEO) as anaesthetic agent in Mozambique tilapia. The main conclusion is that OEO at concentrations of 50 and 100 μL/L meet ideal anaesthetic criteria, but is not safe for inducing anaesthesia in Mozambique tilapia. So, the authors recommended using a lower concentration of 25 μL/L which is safer although it could not induce anaesthesia within 300s (5 min). While the information is important for OEO application in fish, there are some important questions that needed to be answered first.

1.      The common name “tilapia mossambicus” is invalid and is better be replaced by “Mozambique tilapia”. (https://www.fishbase.se/summary/3)

2.      Line 95: Was the OEO used with 100% purity? If not 100% pure, please specify the purity of the OEO.

3.      Line 98-101: The authors should briefly describe the methods of GC/MS and GC/FID as well as their method validation data such as LOD, LOQ, precision, and so on.

4.      Line 105-106: The authors should describe the criteria of ideal anaesthesia in the Materials and Methods. For example, the drug should induce deep anaesthesia within 3-5 min and recover within 5 min. Furthermore, it is unclear why the reference you provided here [Ref#15] is not the same as in Line 131 [Ref#23].

5.      Line 109: Regarding the stages of anaesthesia, why the reference you provided here [Mirghaed et al., Ref#10] (Table 1) is not the same as in Line 124 [Table 1, Ref#22]?

6.      Line 107, 111, and 127: How many fish the authors used in total? From the description, the authors used 9 fish for each concentration. So, there must be a total of 9 fish/conc. x 6 conc. = 54 fish in the first experiment (Line 107-110). Please confirm this information as well as in Line 111 and 127. However, if that is the case, why the authors mentioned: “complete water change after each concentration trial” (Line 114)? Isn’t it normal to complete water change and reset the experiment after finishing each trial if we would like to use the same tank for another set of an experiment? The description in Line 114 made it look like a cross-over design study in which each individual fish is exposed to different chemical concentrations with a proper wash-out period. Please clarify this point.

7.      Line 140: Please clarify what was the time “zero (immediately after anaesthesia induction)”. Whether the authors mean immediately after the completion of 10-min bath in OEO? Or immediately after reaching deep anaesthesia (Stage AD), which was less than 10 min?

8.      Line 157: The blood parameter study was not suitable for 2-way ANOVA analysis as it had only one independent factor (i.e., treatment group). Note that the time “before” and “24 h after” are just time points, not an independent factor. The authors should reanalyze the results using 1-way ANOVA.

9.      Line 177-182 and Figure 1: The authors mentioned the statistical analysis result in Figure 1. However, the results of the 1-way ANOVA analysis were entirely absent from the manuscript.

10.  Line 189-193 and Figure 1: The most problematic part of the whole manuscript is the regression analysis. First, the authors should present the data as average±SD instead of the raw data plotted in Figure 1a and 1b, otherwise, it is very hard to see any relationship between time (y-axis) and concentration (x-axis). Second, Based on the plot in Figure 1c and r2 = 0.31 there was no correlation between full recovery time and deep anaesthesia. Third, it is unclear why the authors chose polynomial regression even if it appeared not to fit with the data. Fourth, where is the statistical analysis result of the regression analysis? Did the result really show a statistically significant correlation at p<0.05? Again, based on Figure 1a-c, they seem to have no or weak correlation unless otherwise indicated by statistical analysis result (p<0.05). To improve the quality of the present manuscript, I suggest the authors remove the regression analysis altogether. The authors can still describe the trend of decreasing induction time as the OEO concentration increased without performing regression analysis.

11.  Line 203-205: The sentence of “Fish bathed in 100 μL/L had significantly higher glucose levels among treatment groups before and after 24 h (p < 0.05), and this level was reduced significantly after 24 h (p <0.05).” was not true because the authors did not perform statistical analysis to compare the glucose level between before and after 24 h within the same treatment group. How can the authors know that the glucose level of the 100 μL/L group was reduced significantly after 24 h (compared to before)?

12.  Line 207-209: The sentence of “Regarding haematological parameters, treatment groups did not significantly affect ..........., except for haemoglobin, lymphocytes, and WBC.” was not true because haemoglobin, lymphocytes, and WBC were also unaffected by the OEO as well. It should be noted that the authors added ethanol to enhance miscibility between OEO and water. So, the ethanol group should be viewed as the real control group (not just water). In this case, no significant differences were detected between the ethanol and the two OEO groups. The more appropriate conclusion is that other than the serum blood glucose no haematological and biochemical parameters were significantly affected by the OEO.

13.  Line 267-268: The sentence of “this may indicate a broad safety limit for OEO as a potential fish anaesthetic, making it easier to apply” was not true. Given that the fish mortality of the 100 μL/L group was 44.44%! on the 7th day following even just a 10-min bath which was deemed unacceptably high, it is likely that OEO may exert a long-term adverse effect on the fish health, thereby making the claim of the safeness and easy-to-use questionable. Apparently, the therapeutic index of OEO was narrow.

Other comments are as follow:

1.      In the title, “Origanum Vulgare” and “Oreochromis Mossambicus” should be replaced by “Origanum vulgare” and “Oreochromis mossambicus”, respectively.

2.      Line 64: “And” should be changed to “and”.

3.      Line 65: “In” should be changed to “in”.

4.      Figure 1c: The unit of the x-axis was absent.

Reviewer 3 Report

The article entitled “The Effects of Origanum Vulgare L Essential Oils on Anaesthesia and Haemato-Biochemical Parameters in Tilapia Mossambicus, Oreochromis Mossambicus Post Juveniles” by Ndakalimwe Naftal Gabriel et al. is interesting and scientifically relevant. The article is well written and easily understandable, materials and methods are clearly explained and conclusions support the results. Authors could improve the introduction by implementing the description of the origin and its effects. I also believe that adding a more critical note to the conclusions may be more impactful for readers. I suggest that the manuscript be accepted for publication after a minor revision.

Round 2

Reviewer 2 Report

Thanks for considering my previous comment. The current manuscript is acceptable after a few minor modifications as follows:
1. Line 204-205: Since the regression analysis was removed from the manuscript, the description of the regression analysis in Line 204-205 should also be deleted.
2. Figure 1: I suggest removing the R2 values in Figure 1.
